# Low use of condom and high STI incidence among men who have sex with men in PrEP programs

**Oskar Ayerdi Aguirrebengoa**[1,2]*, **Mar Vera García**[1], **Daniel Arias Ramírez**[3], **Natalia Gil García**[3], **Teresa Puerta López**[1], **Petunia Clavo Escribano**[1], **Juan Ballesteros Martín**[1], **Clara Lejarraga Cañas**[1], **Nuria Fernandez Piñeiro**[1], **Manuel Enrique Fuentes Ferrer**[4], **Mónica García Lotero**[1], **Estefanía Hurtado Gallegos**[1], **Montserrat Raposo Utrilla**[1], **Vicente Estrada Pérez**[2,5], **Jorge Del Romero Guerrero**[1], **Carmen Rodríguez Martín**[1,2]

1 Centro Sanitario Sandoval, Hospital Clínico San Carlos, IdISSC, Madrid, Spain, 2 Universidad Complutense de Madrid, Madrid, Spain, 3 Medicina Interna, Hospital Clínico San Carlos, IdISSC, Madrid, Spain, 4 Servicio de Medicina Preventiva, IdISSC, Universidad Alfonso X el Sabio, Madrid, Spain, 5 Hospital Fundación Jiménez Díaz, Madrid, Spain

* oskarayerdi@hotmail.com

## Abstract

### Objective

Since the recent introduction of preexposure prophylaxis (PrEP), several studies have reported a decrease in the use of condoms and a rise in STIs among users. This rise in risk behavior associated with the advent of PrEP is known as "risk compensation." The aim of this study is to measure clinical and behavioral changes associated with the introduction of PrEP by analyzing condom use for anal intercourse, number of sexual partners, sexualized drug use and STI incidence.

### Methods

We performed a retrospective descriptive study of PrEP users followed every 3months over a 2-year period spanning 2017–2019 in a referral clinic specializing in STI/HIV in Madrid, Spain. One hundred ten men who have sex with men and transgender women underwent regular screening for STIs and hepatitis C virus (HCV) infection. Sociodemographic, clinical, and behavioral data were gathered for all subjects studied.

### Results

The risk compensation observed in this study consisted primarily of a lower rate of condom use, while the number of sexual partners and recreational drug consumption remained stable. We observed a very high incidence of STIs in this sample, particularly rectal gonorrhea and chlamydia. The factors shown to be independently associated with the presence of an STI on multivariate analysis were age below 30 years and over 10 sexual partners/month.

**Data Availability Statement:** All relevant data are within the manuscript and its Supporting Information files.

**Funding:** The authors received no specific funding for this work.

**Competing interests:** The authors have declared that no competing interests exist.

## Conclusion

The incidence of STI acquisition was higher than expected, indicating a need for strategies to minimize this impact, particularly among younger individuals with a higher number of sexual partners.

## Introduction

HIV preexposure prophylaxis (PrEP) is a preventive measure that consists of administering antiretroviral drugs to uninfected individuals who engage in high-risk sexual behavior in order to avoid infection [1]. Several randomized clinical trials comparing tenofovir disoproxil fumarate (TDF) and emtricitabine (FTC) to a placebo have confirmed that daily oral PrEP is safe and effective [2–4]. The efficacy of PrEP in preventing infection is strongly correlated with levels of adherence [1].

The FDA approved the use of TDF/FTC as PrEP in 2012, and the Centers for Disease Control and Prevention (CDC) have recommended the treatment since 2014 [5]. The IPERGAY trial [6], in which on-demand PrEP was prescribed, before and after sexual activity, showed the same efficacy as the PROUD Study [7], in which PrEP was taken daily. In October 2019, the FDA approved a second drug combination, tenofovir alafenamide (TAF)/FTC, for PrEP in men who have sex with men (MSM) and in transgender women (TGW) [8]. The WHO recommends offering PrEP to people at "substantial" risk of infection who belong to population groups in which the incidence of HIV is over 3 infections per 100 person-years (PY) as well as other preventive measures such as condom use, screening for other STIs, and universal access to early diagnosis and antiretroviral treatment (ART) [9]. The most relevant indications for PrEP use in MSM and TGW are condomless sex with multiple partners, presence of a bacterial STI infection in the rectum, and use of drugs to engage in sexual intercourse [10, 11]. It is believed that PrEP is a cost-effective approach in such cases [12].

Since the introduction of this preventive measure, several studies have reported a decrease in the use of condoms and a rise in STIs among users of PrEP [13]. The concept of exhibiting higher sexual risk after adoption of a safety mesure like PrEP is known as "risk compensation" [14].

In November 2019, the Spanish Ministry of Health announced that it would include PrEP as a publicly funded additional measure of protection against HIV infection within the country's national health system [15]. An increase in PrEP is to be expected in light of this decision, particularly in large cities. Therefore, knowledge of risk compensation among these individuals is essential in order to design specific strategies to limit such compensatory behavior.

The aim of this study is to measure clinical and behavioral changes and risk compensation associated with the advent of PrEP by analyzing condom use for anal intercourse, number of sexual partners, sex-related recreational drug use, and STI incidence.

## Methods

### Study design and study population

We performed a retrospective descriptive study of PrEP users followed over the 2-year period spanning from 2017 to 2019 in a specialized referral clinic for STI/HIV located in Madrid, Spain. A total of 110 MSM and TGW were selected, all of them complete the duration of the study period, which consisted of outpatient visits every three months. On-demand care, was

also provided in cases of clinical or epidemiologic suspicion of STI transmission; all infections diagnosed during these unscheduled visits were addressed in the subsequent screening. The eligibility criteria was taking PrEP, so all participants had sexual risk indications for this preventative measure as proposed by the guidelines [9, 10] and started taking PrEP at the first visit of the study.

## Variables

In the first day visit and at the end of 2-year study period, a structured epidemiological questionnaire was completed systematically to gather sociodemographic, clinical, and behavioral data, which included gender (MSM or TGW), age (20–30; 31–40; >40), region of origin (Spain, Latin America, other), adherence to PrEP (calculated the number of days taking PrEP: high: >90%, low: <90%), condom use before and after PrEP (>50%; <50%, or never), number of sexual partners before and after PrEP (1–5, 6–10, 11–50, >50), substance use (alcohol, cannabis, poppers, cocaine, ecstasy, MDMA, GHB, mephedrone, and methamphetamine), sexualized drug use (condomless sex occurring under the effects of drugs and type), "slamming" (injection of recreational drugs), "chemsex" (sexual activity typically with multiple partners under the effects of drugs), use of substances for erectile enhancement, and dating-app use.

During the 2-year study period, all participants had the following tests performed every 3 months: HIV serology (chemiluminescent micro-particle immunoassay (CMIA) with Western blot confirmation) and syphilis testing (RPR, EIA, and TPPA). Swab-based throat and rectal samples were performed systematically by a health-care professional to detect the following: *Neisseria gonorrhoeae* (NG) by means of Gram staining, Thayer-Martin agar, API NH, and PCR; *Chlamydia trachomatis* (CT) by PCR; lymphogranuloma venereum (LGV) via genotyping; and NG and CT screening of urine samples by PCR. Serology testing for HCV infection (CMIA) was conducted every 6 months. All users of PrEP included were vaccinated against hepatitis A and B infection at the beginning of the study. No patients with acute or chronic hepatitis participated in the study. Follow-up of STIs detected in on-demand visits was included as part of the following scheduled visit.

## Statistical analysis

Qualitative variables are expressed as absolute and relative frequencies. Continuous variables are summarized as mean values and standard deviation (SD) or median and interquartile range (IQR) in case of nonnormal distribution. McNemar's test for paired data was used to compare qualitative variables (frequency of condom use and number of sexual partners) before and after PrEP.

Every 6 months, we calculated the total number of CT and NG diagnoses in the pharynx, rectum, and urethra as well as all cases diagnosed with acute syphilis and HCV infection. We further calculated the incidence rate per 100 person-years (PY) for CT, NG, syphilis, and HCV. Additionally, we determined the overall infection rates (all infection sites combined/any site) by dividing the number of infections by the total time at risk.

A bivariate analysis was performed using the Poisson regression model to identify baseline characteristics related to the rate of incidence of any STI at 2 years. Factors found to be significant on bivariate analysis (p<0.05) were entered into a multivariate Poisson regression model. For the multivariate analysis, related to drug use, the variable sexualized drug was used and not chemsex, as all drugs were accounted for in the former category. Incidence rate ratios are presented alongside their corresponding 95% confidence interval. For all comparisons, the null hypothesis was rejected for a bilateral test of alpha risk of <0.05. Statistical analysis was performed using the STATA statistical software package, release 15.0.

### Ethics statement

Data were obtained from a structured epidemiological questionnaire completed systematically in the course of ordinary clinical practice. All data derived from medical histories were fully anonymized prior to access. The study protocol was approved by the IRB of Hospital Clínico San Carlos, approval Number: 20/214-E. The ethics committee waived the need for informed consent, since the information obtained for the study is collected in routine clinical practice. The study did not include minors.

## Results

All participants in this study were MSM with the exception of two TGW. Mean patient age was 34.7 years (SD: 6.72) with a range of 20 to 60 years. Table 1 contains descriptive variables such as age and region of origin as well as the STIs diagnosed during the enrollment visit, that is, before beginning PrEP.

During the 2-year study period, 98.2% (n = 108) of patients reported high adherence to the drug.

Behavioral changes concerning condom use and number of sexual partners before and after PrEP were analyzed (Table 2). As of the initiation of PrEP, 78.2% (n = 86) of users reported a reduction in condom use for anal intercourse; this decrease was statistically significant (p<0.001). Before PrEP, 85.4% (n = 94) of participants used condoms usually (>50%) in anal intercourse; 10.0% (n = 11) occasionally (<50%) and 4.5% (n = 5) never. After PrEP, the 30.0% (n = 34) of participants used condoms usually, 50.0% (n = 55) occasionally and 20.0%

**Table 1. Descriptive variables for the population studied and STIs detected on enrollment.**

| Variables | % (N) |
|---|---|
| Age | |
| 20–30 | 21.8 (24) |
| 31–40 | 58.2 (64) |
| >40 | 20.0 (22) |
| Region of origin | |
| Spain | 76.4 (84) |
| Latin America | 12.7 (14) |
| Other | 10.9 (12) |
| STIs detected on screening visit | |
| NG | |
| Pharynx | 16.4 (18) |
| Rectum | 22.7 (25) |
| Urethra | 0 (0) |
| CT | |
| Pharynx | 2.7 (3) |
| Rectum | 12.7 (14) |
| Urethra | 1.8 (2) |
| Syphilis | 5.4 (6) |
| Positive hepatitis C virus serology | 0.9 (1) |
| No. of STIs detected on screening visit | |
| 0 | 55.5 (61) |
| 1 | 28.2 (31) |
| 2 | 15.5 (17) |
| 3 | 0.9 (1) |

**Table 2. Behavioral changes concerning condom use and number of sexual partners before and after PrEP.**

|  | Enrollment without PrEP % (N) | After two years on PrEP % (N) |
|---|---|---|
| Use of condom in anal intercourse |  |  |
| • Usually (>50%) | 85.4 (94) | 30.0 (34) |
| • Occasionally (<50%) | 10.0 (11) | 50.0 (55) |
| • Never (0%) | 4.5 (5) | 20.0 (22) |
| Number of sexual partners per month |  |  |
| • 1–5 | 32.7 (36) | 31.8 (35) |
| • 6–10 | 47.3 (52) | 40.0 (44) |
| • 11–50 | 15.65 (17) | 24.5 (27) |
| • >50 | 4.5 (5) | 3.6 (4) |

(n = 22) never. Of the individuals studied, 80.9% reported no increase in the number of sexual partners since beginning PrEP, which reflected no statistically significant change. Before PrEP, 32.7% (n = 36) had 1–5 sexual partners per month, 47.3% (n = 52) 6–10, 15.5% (n = 17) 11–50 and 4.5% (n = 5) more than 50 per month. After PrEP, the 31.8% (n = 35) had 1–5 sexual partners per month, 40.0% (n = 44) 6–10, 24.5% (n = 27) 11–50 and 3.6% (n = 4) more than 50 sexual partners per month.

Alcohol and other recreational drugs were consumed by 94.5% (n = 104) of PrEP users, and 89.1% (n = 98) reported no increased consumption of these substances. Excessive alcohol consumption and use of poppers, GHB, and cocaine were the most common drugs used (Fig 1). While under the effects of these substances, 85.4% (n = 89) engaged in condomless sexual intercourse. The drugs most closely associated with condomless sex were methamphetamine, mephedrone, GHB and poppers. Over half, 53.6% (n = 59), took part in chemsex, with a median number of sessions per year of 4 (IQR: 2–12). Two individuals practiced slamming. Erection-enhancing substances were consumed by 67.3% (n = 74).

Prior to beginning PrEP, 84.5% (n = 93) used dating apps to search for sexual partners, and 55.5% (n = 61) made reference to this on their profile.

No cases of HIV Infection were diagnosed during the study period. The detection rate for STIs of any type was 197.369 cases per 100 PY (Table 3). This table presents the STIs detected

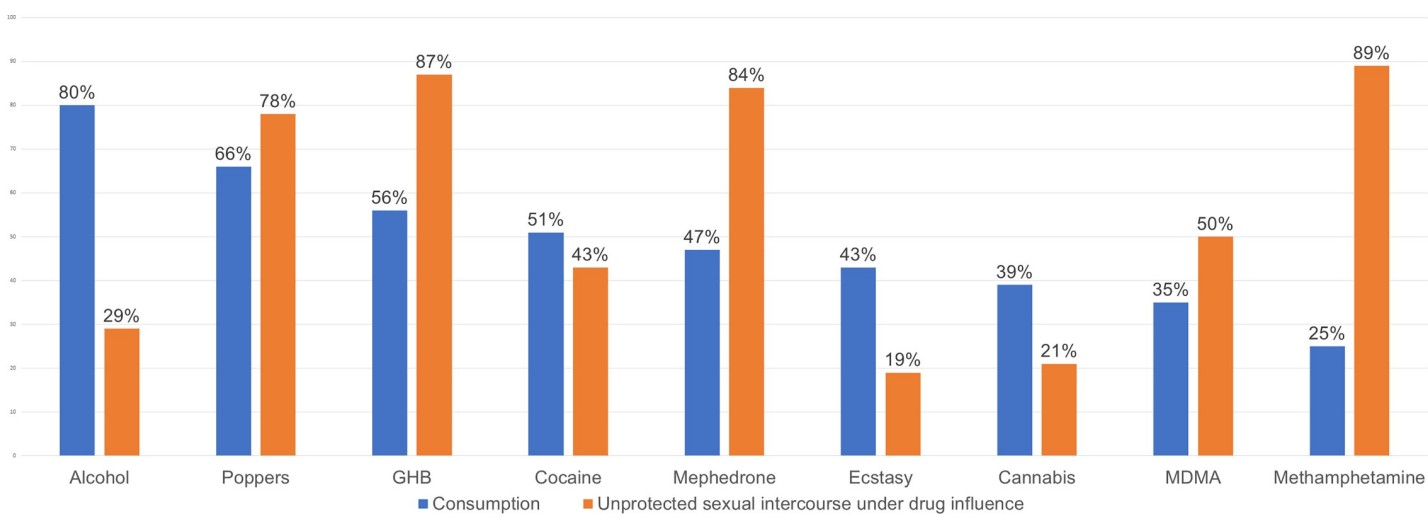

**Fig 1. Frequency of recreational-drug consumption and unprotected sexual intercourse under the influence of these drugs.**

**Table 3. Frequency of STIs among PrEP users during the study period (n = 110).**

|  | No. | Frequency per 100 PY |
|---|---|---|
| NG |  |  |
| Pharynx | 61 | 29.365 (22.46–37.72) |
| Rectum | 174 | 83.761 (71.77–97.17) |
| Urethra | 23 | 11.072 (7.19–16.13) |
| Any location | 219 | 105.423 (91.92–120.35) |
| CT |  |  |
| Pharynx | 13 | 6.258 (3.33–10.70) |
| Rectum | 129 | 62.099 (51.85–73.79) |
| LGV of the rectum | 39 | 18.774 (13.35–25.66) |
| Urethra | 24 | 11.553 (7.40–17.19) |
| Any location | 155 | 74.615 (63.33–87.33) |
| Syphilis |  |  |
| Early latent | 19 | 9.15 (5.51–14.28) |
| Primary | 4 | 1.93 (0.52–4.93) |
| Secondary | 9 | 4.33 (1.98–8.22) |
| Total | 32 | 15.404 (10.54–21.74) |
| Acute hepatitis C | 4 | 1.93 (0.52–4.93) |
| Total STIs, any site | 410 | 197.369 (178.72–217.43) |
| 6-month visit | 106 | 191.751 (156.91–231.91) |
| 12-month visit | 118 | 232.788 (192.68–278.78) |
| 18-month visit | 100 | 194.553 (158.30–236.63) |
| 24-month visit | 86 | 170.770 (136.59–210.90) |
| Number of STIs in any site over 2 years: % (N) |  |  |
| 0 | 5.5 (6) |  |
| 1 | 16.4 (18) |  |
| 2 | 14.5 (16) |  |
| 3 | 12.7 (14) |  |
| 4 | 11.8 (13) |  |
| 5 | 17.3 (19) |  |
| ≥6 | 21.8 (24) |  |

during the 2-year study period as well as the rates of infection by pathogen and by location. The most frequently detected diseases were NG and CT of the rectum. An analysis of the incidence rate at 12, 18, and 24 months revealed no statistically significant changes when compared to the first semester ($IRR_{12m}$: 1.214; p = 0.148; $IRR_{18m}$: 1.015; p = 0.917; $IRR_{24m}$:0.891; p = 0.891).

Table 4 shows the relationship between sociodemographic, clinical and behavioral characteristics and the incidence rate for any STI among the PrEP users studied. On bivariate analysis, the variables associated with a higher rate of incidence were age, number of sexual partners per month of over 10, condomless sex under the influence of drugs (sexualized drug use), participation in chemsex, dating-app use and lower use of condoms for anal sex. A multivariate analysis was performed by including those variables found to be statistically significant in the bivariate analysis (p<0.05). For variables related to drug use, we decided to include the variable sexualized drug use and not chemsex, as all drugs were accounted for in the former category. The factors shown to be independently associated with the presence of an STI on multivariate analysis were age below 30 years and over 10 sexual partners per month.

**Table 4. Demographic, clinical, and behavioral factors associated with STI presence among PrEP users (N = 110).**

| Characteristics | Univariate IRR | p | Multivariate | p |
|---|---|---|---|---|
| Age | | | | |
| Age Continuous | 0.970 (0.95–0.99) | <0.001 | 0.978 (0.96–0.99) | 0.007 |
| 20–30 | 1 | | | |
| 31–40 | 0.956 (0.76–1.21) | 0.700 | | |
| >40 | 0.690 (0.50–0.95) | 0.022 | | |
| Region of origin | | | | |
| Spain | 1 | | | |
| Other | 1.078 (0.86–1.35) | 0.506 | | |
| No. of STIs at enrollment visit, that is, before start of PrEP | | | | |
| 0 | 1 | | | |
| 1 | 0.929 (0.75–1.15) | 0.505 | | |
| 2 | 1.204 (0.89–1.64) | 0.233 | | |
| No. of sexual partners/month before start of PrEP | | | | |
| <10 | 1 | | | |
| >10 | 1.311 (1.05–1.64) | 0.018 | 1.257 (1.00–1.58) | 0.047 |
| Condom use for anal intercourse before start of PrEP | | | | |
| >50% | 1 | | | |
| <50% | 1.284 (1.00–1.64) | 0.046 | 1.247 (0.97–1.61) | 0.086 |
| Condomless sex under the effects of recreational drugs | | | | |
| No | 1 | | | |
| Yes | 1.53 (1.15–2.04) | 0.003 | 1.315 (0.98–1.76) | 0.072 |
| Condomless sex under the effects of recreational drugs, by drug type* | | | | |
| Alcohol | 0.977 (0.78–1.23) | 0.847 | | |
| Cannabis | 1.537 (1.14–2.07) | 0.005 | | |
| Poppers | 1.184 (0.97–1.44) | 0.089 | | |
| Cocaine | 1.321 (1.06–1.64) | 0.012 | | |
| Ecstasy | 1.097 (0.78–1.54) | 0.782 | | |
| MDMA | 1.095 (0.86–1.40) | 0.469 | | |
| GHB | 1.592 (1.30–1.94) | <0.001 | | |
| Mephedrone | 1.543 (1.27–1.87) | <0.001 | | |
| Methamphetamine | 1.453 (1.17–1.80) | 0.001 | | |
| Chemsex | | | | |
| No | 1 | | | |
| Yes | 1.363 (1.12–1.66) | 0.002 | | |
| Dating apps | | | | |
| No | 1 | | | |
| Yes | 1.442 (1.06–1.96) | 0.019 | 1.278 (0.93–1.75) | 0.126 |

* Each of the drugs are evaluated as yes/no and the reference category is "no".

## Discussion

This study evaluated risk compensation among users of PrEP by measuring condom use, number of sexual partners, drug consumption and STI presence. Some existing studies have found no decrease in condom use associated with PrEP [1]. However, among the PrEP users included in the Kaiser cohort, 41% decreased the use of condoms after beginning PrEP [16] as compared with 78% reported here, although we found no independent relationship between condomless sex and an increase in STI incidence.

Although some studies [17] have found an increase in the number of sexual partners, this change did not reach statistical significance for our cohort: 81% had the same number of sexual partners, and a similar rate (74%) was found in a cohort studied in San Francisco [16].

Though chemsex is a common practice among PrEP users, our data reveal no increase in drug use associated with this preventive measure. Although, we found that 15.6% of patients reported missing a dose while under the effects of alcohol or other recreational drugs, this was no significant difference, a finding also reported in the study by O'Halloran et al. [18].

McCormack et al. [6] compared the incidence of STI transmission between users and non-users of PrEP, finding no difference. Nguyen et al. [19] analyzed the rate of STI transmission before and after initiation of PrEP (48 vs. 84 cases per 100 PY) and found a significant increase, partially owing to a greater number of patient visits and STI screening procedures. We observed 197 STI cases per 100 PY, which is substantially above the rate described in most research published to date [13]. The most common infection found in the study by Nguyen et al. [19] was chlamydia (29 cases per 100 PY) followed by rectal gonorrhea and syphilis, both of which had a rate of 15 infections per 100 PY. In our study, the most common infections were rectal gonorrhea (84 cases per 100 PY), rectal chlamydia (62 cases per 100 PY), pharyngeal gonorrhea (29 cases per 100 PY), and rectal LGV (19 cases per 100 PY). These rates are significantly higher than those reported previously, with the exception of syphilis, which was consistent with other reports (15 infections per 100 PY). However, Beymer et al. [20] found that syphilis infection showed the greatest increase after the start of PrEP.

Presence of an STI was most closely related to sexualized drug use followed by a higher number of sexual partners and less frequent condom use. Nonetheless, age under 30 years and over 10 sexual partners per month were the only factors independently associated with the presence of an STI, reaching statistical significance. Young-adult and adolescent MSM and TSW are particularly susceptible to STI/HIV infection [21]. Guidelines could recommend PrEP for adolescents belonging to these population groups, thus making analyses of risk compensation in these individuals particularly beneficial.

Despite the high rate of STI transmission observed, the reduction in condom use was not the only factor involved in the increased rate of infection, as found in other studies [20]. Frequent STI screening in asymptomatic individuals, searching for signs of infection in extragenital sites, many of which show no symptoms, and sample-taking performed by health-care professionals instead of self-testing all facilitate STI detection in cases that would otherwise go unnoticed.

The present study has certain limitations that should be considered. First, it is a retrospective descriptive study conducted in a single specialist center for STI/HIV and includes a small sample size. However, it is the first study of risk compensation in PrEP users carried out in Spain, the number of patients lost to follow-up was very low, and prospective data were recorded meticulously. Other than the results from MSM, the TGW results were not found representative as there were only two participants in the study. In addition to regularly scheduled appointments, several visits were held on demand due to clinical or epidemiologic suspicion of STI transmission. To facilitate data recording and analysis, infections detected during these on-demand visits were addressed in the subsequent scheduled examination, thereby increasing the number of infections per visit. Another limitation of this study is the absence of a control group made up of MSM/TSW who do not use PrEP, in order to compare STI incidence. Future research should include comparisons of cohorts of users and nonusers of PrEP to compare STI incidence and behavioral changes between both groups. This method of preventing HIV infection was included as a publicly funded measure covered under the Spanish national health system, which suggests the potential for additional research in the future.

The risk compensation observed in this study consisted primarily of a lower rate of condom use, while the number of sexual partners and recreational drug consumption remained stable. The incidence of STI acquisition was higher than expected, indicating a need for strategies to minimize this impact, particularly among younger individuals with a higher number of sexual partners.

Data were obtained through a structured epidemiological questionnaire completed systematically filled during the ordinary clinical practice. For this research no specific grant was received from any funding agency in the public, commercial or not-for-profit sectors. All data derived from medical histories were fully anonymized prior to access.

## Supporting information

**S1 Database.**
(XLS)

**S1 File. Variables label.**
(PDF)

## Acknowledgments

Charles Baker, Luisa María Cabello Ballesteros.

## Author Contributions

**Conceptualization:** Oskar Ayerdi Aguirrebengoa, Mar Vera García, Jorge Del Romero Guerrero, Carmen Rodríguez Martín.

**Data curation:** Oskar Ayerdi Aguirrebengoa, Mar Vera García, Daniel Arias Ramírez, Natalia Gil García, Teresa Puerta López, Petunia Clavo Escribano, Juan Ballesteros Martín, Clara Lejarraga Cañas, Nuria Fernandez Piñeiro, Mónica García Lotero, Estefanía Hurtado Gallegos, Montserrat Raposo Utrilla, Jorge Del Romero Guerrero.

**Formal analysis:** Oskar Ayerdi Aguirrebengoa, Manuel Enrique Fuentes Ferrer, Montserrat Raposo Utrilla.

**Investigation:** Oskar Ayerdi Aguirrebengoa, Manuel Enrique Fuentes Ferrer.

**Methodology:** Oskar Ayerdi Aguirrebengoa, Manuel Enrique Fuentes Ferrer, Montserrat Raposo Utrilla.

**Project administration:** Oskar Ayerdi Aguirrebengoa, Mónica García Lotero, Estefanía Hurtado Gallegos, Montserrat Raposo Utrilla.

**Supervision:** Oskar Ayerdi Aguirrebengoa, Vicente Estrada Pérez, Jorge Del Romero Guerrero, Carmen Rodríguez Martín.

**Validation:** Vicente Estrada Pérez, Jorge Del Romero Guerrero, Carmen Rodríguez Martín.

**Visualization:** Mar Vera García, Clara Lejarraga Cañas, Vicente Estrada Pérez, Jorge Del Romero Guerrero, Carmen Rodríguez Martín.

**Writing – original draft:** Oskar Ayerdi Aguirrebengoa, Mar Vera García, Teresa Puerta López, Jorge Del Romero Guerrero, Carmen Rodríguez Martín.

**Writing – review & editing:** Oskar Ayerdi Aguirrebengoa.

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
