## [Decision Letter · Decision Letter 0]

27 Oct 2020

PONE-D-20-27294

Low use of condom and high STI incidence among men who have sex with men and in transgender women in PrEP programs

PLOS ONE

Dear Dr. Ayerdi Aguirrebengoa,

Thank you for submitting your manuscript to PLOS ONE. After careful consideration, we feel that it has merit but does not fully meet PLOS ONE’s publication criteria as it currently stands. Therefore, we invite you to submit a revised version of the manuscript that addresses the points raised during the review process.

Please note that both reviewers raised important questions that should be addressed in a revised version of the manuscript including concerns about overinterpreting results with n=2 TGW (and inclusion of TGW in the title), patterns of PrEP use (daily/on demand), and some of the statistical interpretations. 

We look forward to receiving your revised manuscript.

Kind regards,

J. Gerardo García-Lerma, Ph.D.

Academic Editor

PLOS ONE

Journal Requirements:

3. In the Methods, please discuss whether and how the questionnaire was validated and/or pre-tested. If this did not occur, please provide the rationale for not doing so.

Reviewers' comments:

Reviewer's Responses to Questions

**Comments to the Author**

1. Is the manuscript technically sound, and do the data support the conclusions?

Reviewer #1: No

Reviewer #2: Partly

2. Has the statistical analysis been performed appropriately and rigorously? 

Reviewer #1: Yes

Reviewer #2: Yes

3. Have the authors made all data underlying the findings in their manuscript fully available?

Reviewer #1: Yes

Reviewer #2: Yes

4. Is the manuscript presented in an intelligible fashion and written in standard English?

Reviewer #1: Yes

Reviewer #2: No

5. Review Comments to the Author

Reviewer #1: This manuscript describes clinical and behavioral characteristics of a cohort of PrEP users in Madrid, Spain. The study provides valuable data on STI incidence, sexual risk and substance use behaviors in this group, and this report is especially timely given the recent expansion of public medical coverage in Spain to include PrEP. The regular collection of STI screening data that is clinically verified is an especially valuable contribution. However, there are many methodological details not presented that make interpreting this paper challenging.

Major issues:

1) The title and abstract describe the cohort as “men who have sex with men and transgender women” but the cohort in fact only includes 2 TGW. While there is a clear need for data on STI incidence and behavioral risk factors among TGS PrEP users, I don’t believe it’s appropriate to group them with the rest of this this cohort with the ability to do more detailed sub-group analyses to determine whether the findings are consistent for this population. I believe it would be more appropriate exclude these two participants from this analysis and define the cohort as exclusively cisgender men who have sex with men.

2) The Methods section is currently lacking in sufficient detail to interpret the results and discussion presented. Most critically, I question whether this can be accurately described as a study of risk compensation among PrEP users because there is no control group of non-PrEP users (as mentioned in the limitations), it is not clear whether all participants were already on PrEP at the commencement of the study, and what time frame was being recalled when asked about condom use and substance use before the study. Because high sexual risk and STI infection are indications for PrEP prescription in the first place, this could simply be reporting STI incidence in a cohort of high-risk MSM. Still could be valuable to report, but not in the risk compensation framing.

3) Some details that should be added to the Methods to help interpret this study are as follows:

a. What was the eligibility criteria for participants?

b. Where all participants on PrEP at baseline?

c. What was assessed at baseline and at follow-up surveys? How many follow-up surveys were conducted? Unclear whether it was one at the end of the study or at each of the 3 month screening visits.

d. Adherence to PrEP was assessed as >90% of what? Doses? Over how many days?

e. Condom use before/after PrEP: Time frame for each? What does % refer to? Partners, encounters?

f. Number of partners before/after PrEP: Time frame for each?

g. Time frame/frequency for substance use behaviors? Also, since substance use as defined was near-universal (94.5%), perhaps the authors could consider a different definition of this variable that would have more variability. I assume alcohol use is highly prevalent in this population (as it is in the general population) and not all use is necessarily problematic. The same can be said for drugs like cannabis.

Minor issues:

1) The third paragraph of the Results section could use some copy-editing. Sentence construction like “…the 85.4% (94) use condom usually” is not grammatically correct. Furthermore, the number in parenthesis can be easily confused as a reference number. Would recommend revising to “…85.4% (n=94) of participants used condoms usually” throughout this paragraph.

2) Some improper hyphenation of terms: “recreational-drug use”,

3) Would recommend revision of a few terms:

a. “Qualitative variables” should be “Dichotomous variables”

b. “Quantitative variables” should be “Continuous variables”

c. “Toxic habits” should be “substance use”

d. “univariate analysis” should be “bivariate analysis” (because the model was testing associations between two variables)

4) Some typos identified:

a. “analized” should be “analyzed”

b. “inercourse" should be “intercourse”

c. A couple of instances of “more tan”, which should be “more than”

5) Reference 15 appears to be missing a title.

Reviewer #2: This paper presents an analysis of clinic patients using PrEP from 2017 thru 2019 that looked at the association of STI incidence and condom use, number of sex partners, chemsex. A significant decrease in condom use for anal intercourse was observed after PrEP initiation, but no significant change in number of partners was observed. There were also no changes in recreational drug use observed. This research is important and current as the research on PrEP use and behavioral disinhibition, or risk compensation, is still new and findings have future implications for PrEP implementation.

The manuscript needs to be reviewed for grammar and syntax, preferably by a native English speaker to improve clarity of your prose. Below are my comments and questions.

Minor comments and questions:

-You establish that alcohol and other recreational drugs use are common and 90% reported no increase in the study period. With that said, a lot of data are presented regarding drug use and condomless sex, however it does not seem related to the research question regarding PrEP use. For conciseness, is it possible to omit these findings since they do not relate to, or contribute to answering, the research question?

-The previous comment applies to findings on erection-enhancing substances and using dating apps.

-It would be nice to have a table or figure representing the data on behavioral changes regarding condom use and number of sexual partners before and after PreP was initiated

-Variables are not operationalized the same way. For example, "unprotected sex…" and "condomless sex…" are referring to the same thing. Variable names should be consistent throughout the text and with tables and figures.

Abstract:

-Was statistical significance for the analysis conducted set to 0.10? Methods section indicates only 0.05 threshold. Unless the 0.10 threshold was used for model building (regression analysis), this value should be consistent for significance testing.

Introduction:

-Was the any previous efforts to alleviate risk compensation behaviors in the specialized referral clinic this study took place?

Methods:

-Statistical analysis - count data is quantitative data, not qualitative, is it not?

Results:

- "We found that 15.6% of patients reported missing a dose while under the effects of alcohol or other recreational drugs."

I suggest moving this to the discussion section; this finding seems extraneous and does not contribute to the research question, although it is interesting.

-"Alcohol and other recreational drugs were consumed by 94.5% PrEP users, of whom 89.9% reported no increased consumption of these substances"

Was alcohol and recreational drug use assessed at the follow-up points to examine change in drug use during PrEP use or was change in drug use assessed only at the end?

-"Condomless sex under the influence of recreational drugs" and "Sexualized drug" are referring to the same thing, however the former is only used in the table. I suggest adding this explanation in the text or making it consistent.

-"…changes were compared to the first semester." Because STI incidence/IRR's are being calculated, were the assessment of STI's at each follow-up point ensured to be new infections if STI was reported at baseline or previous follow-up points.

-"Condom use for anal intercourse before start of PrEP" was not significant in the univariate analysis; the confidence interval includes 1 indicating the null hypothesis of no association is not rejected. Please change results and relevant text accordingly.

-Infrequent condom use does not need to be reported as significant at <0.10 if significance threshold is <0.05 for all other analyses.

Discussion:

- "In spite of evidence indicating that engaging in chemsex increases the risk of STI transmission, in our study we observed no significant difference in adherence, a finding also reported in the study by O’Halloran et al. (18)." What is the no difference in adherence referring to? PrEP? And how is the first part (chemsex) of this statement related to this study finding?

-Are there any current or planned future strategize for the specialized referral clinic this study took place?

Tables and figures:

-Was Age and Condomless sex under the influence by drug type analyzed differently than the other variables? A reference category in the univariate analysis is not specified.

-How was the Age variable coded in the multivariate analysis? Based on the results and 0.978 point estimate patients over 30 years old were the reference group?

-Why was Condomless sex under the influence by drug type and Chemsex not included in the multivariate analysis? Chemsex indicates a significant association (as well as some few drugs in the first variable) in the univariate analysis. The rationale for this is found in the results, but I suggest moving to methods section so that readers are aware of this decisions before reading the results/table. Or possibly, you could not present the data not being used in the final report.

6. PLOS authors have the option to publish the peer review history of their article (what does this mean?). If published, this will include your full peer review and any attached files.

Reviewer #1: No

Reviewer #2: **Yes: **Jeffrey S Becasen

---

## [Author Response · Author response to Decision Letter 0]

20 Nov 2020

Dear Editor and Reviewers,

We are very grateful for considering this manuscript for publication and for the suggestions you have made. The review is answered below and we remain available for any additional suggestions.

Journal Requirements:

The information has been obtained through the questions that are systematically asked in the usual clinical practice in the STI/PrEP consultation. The variables collected are detailed in the methodology section. Therefore, a validated questionnaire is not used. In the same way, we have also made other publications previously, for example: Ayerdi Aguirrebengoa O, Vera Garcia M, Rueda Sanchez M, D Elia G, Chavero Méndez B, Alvargonzalez Arrancudiaga M, Bello León S, Puerta López T, Clavo Escribano P, Ballesteros Martín J, Menendez Prieto B, Fuentes ME, García Lotero M, Raposo Utrilla M, Rodríguez Martín C, Del Romero Guerrero J. Risk factors associated with sexually transmitted infections and HIV among adolescents in a reference clinic in Madrid. PLoS One. 2020 Mar 16;15(3):e0228998. doi: 10.1371/journal.pone.0228998. PMID: 32176884; PMCID: PMC7075699.

3. In the Methods, please discuss whether and how the questionnaire was validated and/or pre-tested. If this did not occur, please provide the rationale for not doing so. 

Answered in the previous question.

Updated.

Reviewers' comments:

Reviewer's Responses to Questions

Reviewer #1: This manuscript describes clinical and behavioral characteristics of a cohort of PrEP users in Madrid, Spain. The study provides valuable data on STI incidence, sexual risk and substance use behaviors in this group, and this report is especially timely given the recent expansion of public medical coverage in Spain to include PrEP. The regular collection of STI screening data that is clinically verified is an especially valuable contribution. However, there are many methodological details not presented that make interpreting this paper challenging.

Major issues:

1) The title and abstract describe the cohort as “men who have sex with men and transgender women” but the cohort in fact only includes 2 TGW. While there is a clear need for data on STI incidence and behavioral risk factors among TGS PrEP users, I don’t believe it’s appropriate to group them with the rest of this this cohort with the ability to do more detailed sub-group analyses to determine whether the findings are consistent for this population. I believe it would be more appropriate exclude these two participants from this analysis and define the cohort as exclusively cisgender men who have sex with men.

We agree that the results found in two transgender women can not be representative. However, it is common to find a small percentage of transgender people in PrEP cohorts in developed countries (example reference), so we would prefer not to exclude them from the study. This two participants have presented a very similar behavior so the results of the analysis would be very similar. We fully agree with the reviewer, so we have modified the title, leaving only MSM, and also we have included a section in the discussion commenting on this limitation.

Example Reference: Hoornenborg E, Coyer L, Achterbergh RCA, Matser A, Schim van der Loeff MF, Boyd A, van Duijnhoven YTHP, Bruisten S, Oostvogel P, Davidovich U, Hogewoning A, Prins M, de Vries HJC; Amsterdam PrEP Project team in the HIV Transmission Elimination AMsterdam (H-TEAM) Initiative. Sexual behaviour and incidence of HIV and sexually transmitted infections among men who have sex with men using daily and event-driven pre-exposure prophylaxis in AMPrEP: 2 year results from a demonstration study. Lancet HIV. 2019 Jul;6(7):e447-e455. doi: 10.1016/S2352-3018(19)30136-5. Epub 2019 Jun 6. PMID: 31178284.

Title: Low use of condom and high STI incidence among men who have sex with men in PrEP programs. 

However, if the reviewers consider it necessary to exclude the two participants, 

we are ready to modify it.

2) The Methods section is currently lacking in sufficient detail to interpret the results and discussion presented. Most critically, I question whether this can be accurately described as a study of risk compensation among PrEP users because there is no control group of non-PrEP users (as mentioned in the limitations), it is not clear whether all participants were already on PrEP at the commencement of the study, and what time frame was being recalled when asked about condom use and substance use before the study. Because high sexual risk and STI infection are indications for PrEP prescription in the first place, this could simply be reporting STI incidence in a cohort of high-risk MSM. Still could be valuable to report, but not in the risk compensation framing.

The participants were not taking PrEP on baseline (recruitment visit) and al lof them started with the medication that day. Since then, the presence of STIs has been evaluated during the two-year follow-up. In addition, the variables: condom use, number of sexual partners and drug use, have been compared between the recruitment visit (without PrEP), and second year follow up visit (after two years on PrEP) to assess risk compensation.

We remain available for any further clarification. 

3) Some details that should be added to the Methods to help interpret this study are as follows:

a. What was the eligibility criteria for participants? Updated. 

b. Where all participants on PrEP at baseline? Updated.

c. What was assessed at baseline and at follow-up surveys? How many follow-up surveys were conducted? Unclear whether it was one at the end of the study or at each of the 3 month screening visits. Updated, in the firts visit and at the end. 

d. Adherence to PrEP was assessed as >90% of what? Doses? Over how many days? The adherence of the PrEP was calculated by dose and days. Adherence above 90% was considered when they took more than 81 tablets every 3 months (90 days). Updated.

e. Condom use before/after PrEP: Time frame for each? What does % refer to? Partners, encounters? Updated. Table 2. 

f. Number of partners before/after PrEP: Time frame for each? Updated. Table 2.

g. Time frame/frequency for substance use behaviors? Also, since substance use as defined was near-universal (94.5%), perhaps the authors could consider a different definition of this variable that would have more variability. I assume alcohol use is highly prevalent in this population (as it is in the general population) and not all use is necessarily problematic. The same can be said for drugs like cannabis. 

The substance use was analyzed the first day and two years follow up visit. This is the reason, we used “sexualized drugs” variable for the statistical analized. 

Minor issues:

1) The third paragraph of the Results section could use some copy-editing. Sentence construction like “…the 85.4% (94) use condom usually” is not grammatically correct. Furthermore, the number in parenthesis can be easily confused as a reference number. Would recommend revising to “…85.4% (n=94) of participants used condoms usually” throughout this paragraph. Updated.

2) Some improper hyphenation of terms: “recreational-drug use”, Updated.

3) Would recommend revision of a few terms:

a. “Qualitative variables” should be “Dichotomous variables”. We prefer to keep the term “qualitative variables” since it includes both: dichotomous (yes/no) and polytomous (condom use…) variables. 

b. “Quantitative variables” should be “Continuous variables”. Updated.

c. “Toxic habits” should be “substance use”: Updated

d. “univariate analysis” should be “bivariate analysis” (because the model was testing associations between two variables) Updated.

4) Some typos identified:

a. “analized” should be “analyzed”: Updated

b. “inercourse" should be “intercourse”: Updated

c. A couple of instances of “more tan”, which should be “more than”: Updated

5) Reference 15 appears to be missing a title.Updated, Press reléase.

Reviewer #2: This paper presents an analysis of clinic patients using PrEP from 2017 thru 2019 that looked at the association of STI incidence and condom use, number of sex partners, chemsex. A significant decrease in condom use for anal intercourse was observed after PrEP initiation, but no significant change in number of partners was observed. There were also no changes in recreational drug use observed. This research is important and current as the research on PrEP use and behavioral disinhibition, or risk compensation, is still new and findings have future implications for PrEP implementation.

The manuscript needs to be reviewed for grammar and syntax, preferably by a native English speaker to improve clarity of your prose. Below are my comments and questions.

Minor comments and questions:

-You establish that alcohol and other recreational drugs use are common and 90% reported no increase in the study period. With that said, a lot of data are presented regarding drug use and condomless sex, however it does not seem related to the research question regarding PrEP use. For conciseness, is it possible to omit these findings since they do not relate to, or contribute to answering, the research question?

We agree that the information provided on drugs is not entirely necessary to answer the research question. However, there are not so many studies that analyze drug use among PrEP users (none in Spanish population), so we consider is a relevant information, which if it seems right to you, we would prefer to keep.

-The previous comment applies to findings on erection-enhancing substances and using dating apps. Same answer as the previous question.

-It would be nice to have a table or figure representing the data on behavioral changes regarding condom use and number of sexual partners before and after PreP was initiated

Table 2 included. 

-Variables are not operationalized the same way. For example, "unprotected sex…" and "condomless sex…" are referring to the same thing. Variable names should be consistent throughout the text and with tables and figures. Updated.

Abstract:

-Was statistical significance for the analysis conducted set to 0.10? Methods section indicates only 0.05 threshold. Unless the 0.10 threshold was used for model building (regression analysis), this value should be consistent for significance testing.

No, statistical significance value was 0.05. Updated.

Introduction:

-Was the any previous efforts to alleviate risk compensation behaviors in the specialized referral clinic this study took place. 

The preventive advice for safe sex is offered to all PrEP users, as indicated in PrEP guidelines.

Methods:

-Statistical analysis - count data is quantitative data, not qualitative, is it not? Statistical analysis updated. 

Results:

- "We found that 15.6% of patients reported missing a dose while under the effects of alcohol or other recreational drugs." I suggest moving this to the discussion section; this finding seems extraneous and does not contribute to the research question, although it is interesting.

Updated. 

-"Alcohol and other recreational drugs were consumed by 94.5% PrEP users, of whom 89.9% reported no increased consumption of these substances". Was alcohol and recreational drug use assessed at the follow-up points to examine change in drug use during PrEP use or was change in drug use assessed only at the end?

Just at the end. 

-"Condomless sex under the influence of recreational drugs" and "Sexualized drug" are referring to the same thing, however the former is only used in the table. I suggest adding this explanation in the text or making it consistent.

Updated. Added in the text. 

-"…changes were compared to the first semester." Because STI incidence/IRR's are being calculated, were the assessment of STI's at each follow-up point ensured to be new infections if STI was reported at baseline or previous follow-up points.

On baseline we just have the new STI detected in enrollment visit and we do not have previous STI data. So to comparing STI incidence, we calculate it from first 6 months period. Table 3. All of them were new STI detected during the study. 

-"Condom use for anal intercourse before start of PrEP" was not significant in the univariate analysis; the confidence interval includes 1 indicating the null hypothesis of no association is not rejected. Please change results and relevant text accordingly.

To avoid an excessive number of data in table 4, the confidence interval has been rounded to two decimal places. The real confidence interval is (1.004129-1.644366) and therefore does not include 1.

-Infrequent condom use does not need to be reported as significant at <0.10 if significance threshold is <0.05 for all other analyses. Updated. 

Discussion:

- "In spite of evidence indicating that engaging in chemsex increases the risk of STI transmission, in our study we observed no significant difference in adherence, a finding also reported in the study by O’Halloran et al. (18)." What is the no difference in adherence referring to? PrEP? And how is the first part (chemsex) of this statement related to this study finding? Error, updated. 

-Are there any current or planned future strategize for the specialized referral clinic this study took place? Intensify preventive advice especially among younger PrEP users.

Tables and figures:

-Was Age and Condomless sex under the influence by drug type analyzed differently than the other variables? A reference category in the univariate analysis is not specified.

Updated. Included for age and explained with a note in table 4 for “Condomless sex under the effects of recreational drugs, by drug type”.

-How was the Age variable coded in the multivariate analysis? Based on the results and 0.978 point estimate patients over 30 years old were the reference group?

Updated. 

-Why was Condomless sex under the influence by drug type and Chemsex not included in the multivariate analysis? Chemsex indicates a significant association (as well as some few drugs in the first variable) in the univariate analysis. The rationale for this is found in the results, but I suggest moving to methods section so that readers are aware of this decisions before reading the results/table. Or possibly, you could not present the data not being used in the final report. 

Updated. 

Kind regards,

Oskar Ayerdi

---

## [Decision Letter · Decision Letter 1]

18 Dec 2020

PONE-D-20-27294R1

Low use of condom and high STI incidence among men who have sex with men in PrEP programs

PLOS ONE

Dear Dr. Ayerdi Aguirrebengoa,

Thank you for submitting your manuscript to PLOS ONE. After careful consideration, we feel that it has merit but does not fully meet PLOS ONE’s publication criteria as it currently stands. Therefore, we invite you to submit a revised version of the manuscript that addresses the points raised during the review process.

Although much improved, one of the reviewers still has some minor comments that need to be addressed before we can consider it for publication. 

We look forward to receiving your revised manuscript.

Kind regards,

J. Gerardo García-Lerma, Ph.D.

Academic Editor

PLOS ONE

Reviewers' comments:

Reviewer's Responses to Questions

**Comments to the Author**

1. If the authors have adequately addressed your comments raised in a previous round of review and you feel that this manuscript is now acceptable for publication, you may indicate that here to bypass the “Comments to the Author” section, enter your conflict of interest statement in the “Confidential to Editor” section, and submit your "Accept" recommendation.

Reviewer #1: (No Response)

2. Is the manuscript technically sound, and do the data support the conclusions?

Reviewer #1: Yes

3. Has the statistical analysis been performed appropriately and rigorously? 

Reviewer #1: Yes

4. Have the authors made all data underlying the findings in their manuscript fully available?

Reviewer #1: No

5. Is the manuscript presented in an intelligible fashion and written in standard English?

Reviewer #1: No

6. Review Comments to the Author

Reviewer #1: Overall, I find this manuscript to be much improved following revisions. Aside from a few more minor comments, enumerated below, I find this manuscript to be publishable.

Minor comments:

1. Did any participants discontinue PrEP over the course of the follow-up? Or was the sample selected to include only those who stayed on PrEP for the duration of the study period? Either way, please include this information in the Methods section.

2. Table 3 mentions "any distant metastases" under "CT" but this was not included in the Methods section.

3. This manuscript would benefit for one more round of copy-editing for English grammar/style, especially in the Methods and Results sections. Some suggested revisions:

Methods:

- The sentence about eligibility criteria should read: "The eligibility criteria was taking PrEP, so all participants had sexual risk indications for this preventative measure as proposed by the guidelines [reference? More clarity on which guidelines needed here] and started taking PrEP at the first visit of the study."

- "sexualized drug use" instead of just "sexualized drugs"

Results:

- "the 85.4% (n=94) of participants" should just read "85.4% (n=94) of participants". There are a few other instances of this sentence structure in the same paragraph, which should be revised similarly.

- "...consumed by 94.5% of PrEP users" (missing the word of)

- some inconsistency with how % are reported, i.e. sometimes with (n=) and sometimes without. Please revise such that all are consistent and according to journal standards.

- "It is also determined the number of diagnosed STIs per individual." - I don't understand this sentence and it is not clear what result this is reporting.

7. PLOS authors have the option to publish the peer review history of their article (what does this mean?). If published, this will include your full peer review and any attached files.

Reviewer #1: **Yes: **Maria Zlotorzynska

---

## [Author Response · Author response to Decision Letter 1]

4 Jan 2021

PONE-D-20-27294R1

Low use of condom and high STI incidence among men who have sex with men in PrEP programs

Dear Editor,

We are very grateful for the feedback we have received. All the comments have been taken into consideration and we revised the manuscript accordingly. Responses to such comments point-by-point are sent below. We attached revised manuscript in two versions: one clean revised version and a version with track changes.

We remain at your disposal for any additional clarification.

Kind regards

Minor comments:

1. Did any participants discontinue PrEP over the course of the follow-up? Or was the sample selected to include only those who stayed on PrEP for the duration of the study period? Either way, please include this information in the Methods section. All participants selected stayed on PrEP during study period. Updated in methods section. 

2. Table 3 mentions "any distant metastases" under "CT" but this was not included in the Methods section. Mistake updated, we would mean “Any location”. 

3. This manuscript would benefit for one more round of copy-editing for English grammar/style, especially in the Methods and Results sections. Some suggested revisions:

Methods:

- The sentence about eligibility criteria should read: "The eligibility criteria was taking PrEP, so all participants had sexual risk indications for this preventative measure as proposed by the guidelines [reference? More clarity on which guidelines needed here] and started taking PrEP at the first visit of the study." Updated.

- "sexualized drug use" instead of just "sexualized drugs" Updated. 

Results:

- "the 85.4% (n=94) of participants" should just read "85.4% (n=94) of participants". There are a few other instances of this sentence structure in the same paragraph, which should be revised similarly. Updated.

- "...consumed by 94.5% of PrEP users" (missing the word of) Updated.

- some inconsistency with how % are reported, i.e. sometimes with (n=) and sometimes without. Please revise such that all are consistent and according to journal standards. Updated. 

- "It is also determined the number of diagnosed STIs per individual." - I don't understand this sentence and it is not clear what result this is reporting. 

We wanted to make references to the number of STIs diagnosed, but this information is clear enough in Table 3, so this sentence has been removed from the results.

---

## [Editor Report · Decision Letter 2]

11 Jan 2021

Low use of condom and high STI incidence among men who have sex with men in PrEP programs

PONE-D-20-27294R2

Dear Dr. Ayerdi Aguirrebengoa,

We’re pleased to inform you that your manuscript has been judged scientifically suitable for publication and will be formally accepted for publication once it meets all outstanding technical requirements.

Kind regards,

J. Gerardo García-Lerma, Ph.D.

Academic Editor

PLOS ONE
---

## [Editor Report · Acceptance letter]

25 Jan 2021

PONE-D-20-27294R2 

Low Use of Condom and High STI Incidence among men who have sex with men In PrEP Programs 

Dear Dr. Ayerdi Aguirrebengoa:

I'm pleased to inform you that your manuscript has been deemed suitable for publication in PLOS ONE. Congratulations! Your manuscript is now with our production department. 

Kind regards, 

on behalf of

Dr. J. Gerardo García-Lerma 

Academic Editor

PLOS ONE